# Paleogenomes Reveal a Complex Evolutionary History of Late Pleistocene Bison in Northeastern China

**DOI:** 10.3390/genes13101684

**Published:** 2022-09-20

**Authors:** Xindong Hou, Jian Zhao, Hucai Zhang, Michaela Preick, Jiaming Hu, Bo Xiao, Linying Wang, Miaoxuan Deng, Sizhao Liu, Fengqin Chang, Guilian Sheng, Xulong Lai, Michael Hofreiter, Junxia Yuan

**Affiliations:** 1State Key Laboratory of Biogeology and Environmental Geology, China University of Geosciences, Wuhan 430078, China; 2School of Environmental Studies, China University of Geosciences, Wuhan 430078, China; 3Institute for Ecological Research and Pollution Control of Plateau Lakes, School of Ecology and Environmental Science, Yunnan University, Kunming 650500, China; 4Institute for Biochemistry and Biology, University of Potsdam, Karl-Liebknecht-Strasse 24–25, 14476 Potsdam, Germany; 5School of Earth Sciences, China University of Geosciences, Wuhan 430074, China; 6Faculty of Materials Science and Chemistry, China University of Geosciences, Wuhan 430078, China; 7Department of Scientific Research, Dalian Natural History Museum, Dalian 116023, China

**Keywords:** ancient DNA, most recent common ancestor, fossil, genetic diversity, admixture

## Abstract

Steppe bison are a typical representative of the Mid-Late Pleistocene steppes of the northern hemisphere. Despite the abundance of fossil remains, many questions related to their genetic diversity, population structure and dispersal route are still elusive. Here, we present both near-complete and partial mitochondrial genomes, as well as a partial nuclear genome from fossil bison samples excavated from Late Pleistocene strata in northeastern China. Maximum-likelihood and Bayesian trees both suggest the bison clade are divided into three maternal haplogroups (A, B and C), and Chinese individuals fall in two of them. Bayesian analysis shows that the split between haplogroup C and the ancestor of haplogroups A and B dates at 326 ky BP (95% HPD: 397-264 ky BP). In addition, our nuclear phylogenomic tree also supports a basal position for the individual carrying haplogroup C. Admixture analyses suggest that CADG467 (haplogroup C) has a similar genetic structure to steppe bison from Siberia (haplogroup B). Our new findings indicate that the genetic diversity of Pleistocene bison was probably even higher than previously thought and that northeastern Chinese populations of several mammalian species, including Pleistocene bison, were genetically distinct.

## 1. Introduction

During the Pleistocene, the genus *Bison* was widely distributed in Holarctic regions and evolved into different species, adapted to various environmental conditions [1,2,3,4]. Nowadays, there are only two extant species, the European bison (*Bison bonasus*) and the American bison (*Bison bison*), confined to relatively limited regions [5,6,7]. Among the recently extinct species, the steppe bison (*Bison priscus*) was the most wide-ranging one, inhabiting the Mid-Late Pleistocene steppes of the northern hemisphere. Its habitat ranged from Europe across Asia to North America, which is also called the great Pleistocene bison belt [8,9,10,11,12]. However, despite its abundant remains, this extinct species is still relatively poorly known regarding its origin, dispersal and population divergence [1,12,13].

In recent years, many genetic studies have been performed on steppe bison remains from Europe, Siberia and North America [3,10,14,15,16,17]. Earlier maternal phylogenetic analyses indicated that the analyzed specimens represented a relatively homogeneous population, that is, all samples analyzed clustered into the main bison mitochondrial clade [10,18]. This situation changed with the discovery of a frozen early Holocene bison carcass (9497 ± 92 14C yr BP) from the mouth of the Rauchua River, Chukotka, Russia (69°30′ N, 166°49′ E) [13,19]. Analysis of its mitochondrial DNA revealed that this individual fell outside the known genetic diversity of steppe bison, forming a deeply divergent sister lineage to all other bison mitochondrial sequences [19], hereafter referred to as bison haplogroup C. Surprisingly, further investigation of mitogenomes from 26 additional bison samples collected from northeastern Siberia, including sites at Rauchua River, Middle Indigirka River and the Bilibinsky region, showed that all the newly analyzed samples fell into the main Eurasian steppe bison clade rather than haplogroup C [17]. Hence, the Pleistocene to Holocene distribution of bison haplogroup C has remained unknown and its geographical origin is still unclear. It has been suggested that bison carrying haplogroup C likely dispersed northward from their original habitat and escaped from the Late Pleistocene extinction event in the northern Siberian refuge [17], maybe similar to the Holocene survival of the giant deer (*Megaloceros giganteus* B.) [20,21]. However, so far it remains an open question where this lineage originated from and how and when its distribution changed. Situated to the south of Rauchua, northeastern China is an obvious candidate region for the geographical origin of this enigmatic bison lineage.

Northeastern China is one of the classic regions for the Late Pleistocene “*Mammuthus**-Coelodonta* Fauna”, in which members of the family Bovidae were common [22,23,24,25]. According to the fossil record, at least three genera of Bovidae (i.e., *Bubalus*, *Bos* and *Bison*) coexisted in this region during that time, including several named species, e.g., *Bubalus wansjocki*, *Bubalus teilbardi*, *Bubalus brevicornnis*, *Bos primigenius*, *B. priscus*, *Bison exiguus* and several others [22,23,26,27,28]. Especially the members of the genus Bison dominated the local fauna during the Late Pleistocene. Fossil bison in northeastern China were initially identified as steppe bison (*B. priscus*) based on morphological analyses [27,29], while other scholars argued that it should be assigned to a separate species, *B. exiguus*, which was further divided into three subspecies, i.e., *Bison exiguus exiguus*, *Bison exiguus curvicornis* and *Bison exiguus harbinensis* [22,23,28]. So far, the taxonomic assignment of northeastern Chinese bison has remained controversial. Ancient DNA provides a powerful tool for revealing the evolution of extinct populations [30,31,32]. Unfortunately, to our knowledge, only two partial D-loop sequences were retrieved from fossil bison excavated from China so far [10]. Thus, almost no genetic information on Chinese fossil bison is available.

In this study, we generated near-complete and partial mitochondrial genomes from Late Pleistocene bison collected from northeastern China. A low coverage nuclear genome was also retrieved from our best-preserved bison fossil. Using the obtained sequences as well as previously published data, we performed phylogenetic analyses, principal component analyses (PCA) and admixture analyses to explore their relationships with their counterparts from Eurasia and America. We also estimated the divergence time and genetic distance of different bison lineages. Our results suggest a complex history of extinct bison lineages in northeastern China and contribute to filling the gap in understanding the evolution of *Bison* during the Late Pleistocene and early Holocene epochs in Asia.

## 2. Materials and Methods

### 2.1. Sampling Information

In total, 21 fossil bison specimens from northeastern China were included in this study; 19 samples were excavated from Heilongjiang province (Haerbin, Zhaodong, Zhaoyuan, Qinggang and Daqing sites), and two samples from Dalian, Liaoning province (Figure 1, Appendix A). Accelerator mass spectrometry (AMS)-radiocarbon dating was performed on two of these specimens (CADG456 and CADG467) at the β Analytic inc. in the USA (Lab Nos: β-609039 and β-583723). Detailed information on the samples is listed in Appendix A. The base map of the global world (chart no. GS (2021) 5443) was downloaded from the National Administration of Surveying, Mapping and Geoinformation of China (http://bzdt.ch.mnr.gov.cn, accessed on 22 March 2022).

### 2.2. DNA Extraction, Double-Stranded Library Construction and Sequencing

We performed DNA extraction and library preparation on 19 bison samples (Appendix A) in a dedicated ancient DNA laboratory at the China University of Geosciences (Wuhan). DNA extraction was performed following the protocol described in Hu et al. [33]. Each specimen was ground into about 200 mg fine powder with a mortar and pestle. The bone powder was digested in 4.5 mL of EDTA (0.5 M, pH = 8) and 0.06 mL of Proteinase K (20 mg/mL) by incubation in a hybridization oven at 37 °C under constant agitation for 16 h. After centrifugation at 7000 rpm for 10 min, we recovered the supernatant with an ultrafiltration tube (Millipore, Germany) and concentrated it to 100 µL at 7000 rpm for 35 min. Finally, we purified and eluted DNA into 80 µL EB buffer using the QIAquick PCR Purification Kit (Qiagen, Germany) following the manufacturer’s instructions. To monitor potential contamination, we set up blank controls in both experimental steps.

Multiple double-stranded libraries for Illumina high-throughput sequencing were constructed from 20 µL DNA extract according to the protocol described by Meyer and Kircher [34]. In the blunt-end repair step, DNA templates were mixed with BSA (New England Biolabs, Ipswich, MA, USA) and NEB buffer 2 (New England Biolabs, Ipswich, MA, USA). After that, we ligated a 1:20 adapter dilution (Sangon Biotech, Shanghai, China) and the ends of DNA fragments using Quick Ligase Buffer (New England Biolabs, Ipswich, MA, USA) and Quick Ligase (New England Biolabs, Ipswich, MA, USA). The ligation reaction was then added to an adapter fill-in reaction. Indexing PCR amplifications were carried out with Q5 Hot Start High-Fidelity 2 × Master Mix (New England Biolabs, Ipswich, MA, USA) under the following conditions: 98 °C for 30 s and 17 cycles of 98 °C for 10 s, 60 °C for 75 s, and 60 °C for 6 min. Quantitative analyses of the libraries were conducted with Qubit 4.0 (Invitrogen, Carlsbad, CA, USA) and TapeStation 4150 (Agilent, Santa Clara, CA, USA). After pooling 10 or 40 libraries in equimolar ratios, calculated from concentration and DNA length, the libraries were finally sequenced on an Illumina HiSeq×10 platform at Annoroad Inc., Beijing, China.

### 2.3. Single-Stranded Library Construction and Hybridization Capture

In addition, we retrieved mitogenomes from two bison samples (CADG456 and CADG471) (Appendix A) in a dedicated DNA laboratory at the University of Potsdam. DNA extraction was performed as previously published by Sheng et al. [35]. Then, we prepared single-stranded DNA libraries using 20 μL DNA extract of each sample following the protocol described by Gansauge and Meyer [36] with slight modifications as in Yuan et al. [37]. Hybridization capture of the complete mitochondrial genome was carried out using the procedures described in Gonzalez-Fortes and Paijmans [38]. We prepared baits using a modern cattle sample with the following steps: Firstly, four overlapping long-range PCR (LR-PCR) primer pairs (Appendix A) were used to amplify the mitochondrial genome. Secondly, the amplified modern cattle mitochondrial DNA fragments were sheared, blunt-end repaired and ligated to biotinylated adapters. Thirdly, two rounds of hybridization capture were carried out to improve the enrichment rate by the protocol of Yuan et al. [37]. Finally, the enriched libraries were pooled into a single pool in equimolar ratios and sequenced on the Illumina NextSeq 500 sequencing platform in 75 bp single-end runs, following Paijmans et al. [39].

### 2.4. Sequence Assembly

Cutadapt v1.4.2 [40] was used to trim adapters. After discarding reads shorter than 30 bp, the overlapping read pairs were merged with Flash v1.2.11 [41]. The trimmed reads were mapped to four bison complete mitochondrial sequences (GenBank Nos. KR350472, KM593920, KX269145 and GU946980) and the nuclear genome assembly of a modern American bison (GenBank No. GCA_018282365.1) processed with the “aln” and “samse” algorithms in Burrows-Wheeler Aligner v0.6.2 [42] with default parameters. Sequences with a map quality score less than 30 were removed by using “view” and the alignment was sorted on the reference genome by a 5′ mapping position using “sort” in SAMtools v0.1.19 [43]. After removing all potential PCR duplicates by using “rmdup”, the resulting bam files for each library were merged into a single bam file using “merge” from SAMtools. Then the merged bam file was imported to Qualimap v2.2.1 [44] and mapDamage v2.0 [45] to calculate read coverage and test the authenticity of ancient DNA, respectively (Appendix A, Appendix A). Finally, the mitochondrial consensus sequence was called with a minimum coverage of 2× and a base agreement greater than 75% and exported to Geneious v.10 (http://www.geneious.com/, accessed on 7 January 2022).

Identical methods were used to assemble mitochondrial genomes or nuclear genome data from short read data of two ancient steppe bison (875 and 3133), three aurochs, six American bison, ten European bison and two yaks downloaded from ENA (the European Nucleotide Archive) (Appendix A).

### 2.5. Mitochondrial Phylogenetic Analyses

In order to investigate the phylogenetic status of the Chinese bison mitochondrial haplotypes, we performed a maximum-likelihood (ML) phylogenetic analysis with RAxML-HPC v8 [46] on the CIPRES server [47]. Three newly obtained mitochondrial genomes (CADG456, CADG465 and CADG467) were used to reconstruct the phylogenetic tree, which was aligned with 59 steppe bison, ten American bison and four yak sequences downloaded from GenBank and ENA (Appendix A) using MAFFT v7.471 [48]. After removing the 891 bp long D-loop region that cannot be aligned properly, a 15,431 bp alignment was generated based on those positions where bases could be called with confidence. Considering the heterogeneity of evolutionary rates among partitions, one GTR, one GTR + G and two GTR + I nucleotide substitution models suitable for four partitions were generated by PartitionFinder v2.1.1 [49] with the greedy search algorithm, linked branch lengths, and the Bayesian Information Criterion. Using 500 bootstrap replicates, node support was also calculated (Figure 2a).

To infer the divergence times of the different bison haplogroups, we constructed a time-tree with BEAST v1.8.3 [50] using complete mitochondrial genomes. We discarded sequences lacking accurate dating information and performed a Bayesian analysis including only one of our newly obtained mitochondrial genomes (CADG467) plus 32 steppe bison and ten American bison (Appendix A). We used the time of the most recent common ancestor (TMRCA) of bison from America (165 ky BP, 95% HPD: 195-135 ky BP) [14] to calibrate the phylogeny. Tip dates of the samples were also used as calibration points using either the median calibrated radiocarbon ages or stratigraphic ages. The best fitting nucleotide substitution model (HKY + I) was inferred by jModelTest v2.1.6 [51]. We ran the Markov Chain-Monte Carlo (MCMC) chain for 60 million iterations, sampling priors and trees every 1000 iterations. Then, the log file was visually inspected using Tracer v1.7 [52] to ensure all parameters had reached effective sample sizes (ESS) above 200. A maximum clade credibility tree was calculated on the tree file using TreeAnnotator v1.8.0 [50] after discarding the first 10% of sampled states as burn-in. The final tree was visualized and annotated in FigTree v1.4.3 (http://tree.bio.ed.ac.uk/software/figtree, accessed on 8 January 2022).

Following the phylogenetic analyses by Shapiro et al. [10], we also used a dataset that only contains 639 bp of the D-loop region and constructed another phylogenetic tree by Bayesian inference in MRBAYES v3.2.7 [53]. The alignment comprises our two newly obtained sequences (CADG456 and CADG467), 95 partial D-loop sequences described by Shapiro et al. [10] (Appendix A) and 67 complete mitochondrial genomes downloaded from GenBank (Appendix A). The obtained D-loop sequence of the sample CADG 465 is very short, hence it was excluded in this analysis. All other parameters were the same as in the previous Bayesian analysis.

### 2.6. Nuclear Analyses

A total number of 24 individuals were used in principal component analysis (PCA) and admixture proportion estimation (NGSadmix), which contains our best-preserved sample (CADG467) and 23 bovid genomes downloaded from ENA (Appendix A). We only considered the autosomal chromosomes to avoid different rates of evolution and inheritance. Genotype likelihood estimations (GL) were obtained using angsd v0.921 [54] with the following flags: minimum base quality (-minQ 30), minimum mapping quality (-minMapQ 30), infer major and minor alleles from GL (-doMajorMinor 1), remove all reads showing multiple hits (-uniqueOnly 1), only consider SNPs with *p*-value below (-SNP_pval 1e-6), remove transitions (-rmtrans 1), calculate site allele frequencies (-doMaf 1), minimum minor allele frequency (-minmaf 0.0476), skip triallelic sites (-skiptriallelic 1), only consider sites where all steppe bison have coverage (-minind 21), only include scaffolds above 1 MB (-rf). The downstream PCA and NGSadmix analyses are based on the GL estimations. PCAngsd v0.98 [55] was used to construct a covariance matrix on the obtained GL beagle file. Individual admixture proportions were calculated using NGSadmix v32 [56], specifying K = 2–6. To evaluate convergence in our results for each K, we ran each calculation 100 times with an independent seed value. If the best likelihood could be repeated twice, then the result was considered to have converged and thus be reliable. In addition, we later performed another GL estimation for PCA with only nine individuals, i.e., our one sample CADG467, two steppe bison and six American bison.

In order to investigate the nuclear phylogenetic position of our sample, we constructed a rooted nuclear neighbor-joining (NJ) tree on the same data set that was used for the first GL estimation which is based on a total of 24 individuals (Appendix A). The distance matrix was calculated by using the same flags as above in the GL estimation except for adding the filter -doIBS 1 -makeMatrix. Then the NJ tree was calculated using the R package ape [57].

## 3. Result

In total, 21 bison individuals excavated from northeastern China were included in this study, DNA was obtained from six individuals (CADG456, CADG465, CADG467, CADG471, CADG605 and CADG627) (Appendix A). Five of these six samples, excluding CADG465, show a higher coverage when mapping against the mitogenome of the Rauchua bison (GenBank No. KR350472) than those of other steppe bison (GenBank No. KM593920 and KX269145) and American bison (GenBank No. GU946980) (Appendix A). Considering coverage issues, only three samples (CADG456, CADG465 and CADG467), which show a mean coverage of 9.1-fold, 2.2-fold and 8.3-fold, respectively, when mapping against the consensus mitogenome of the Rauchua bison (GenBank No. KR350472), were included in downstream analyses. The mitochondrial DNA fragments are short (average length: 51 bp for CADG456, 47 bp for CADG465 and 63 bp for CADG467), and CADG456 and CADG465 show elevated C to T substitution rates at the 5′ ends and elevated G to A substitution rates at the 3′ ends, in agreement with the damage characteristics expected for double stranded libraries constructed from ancient DNA (Appendix A).

For the analysis of the nuclear sequences, we used one sample (CADG467) with sufficient coverage and 23 previously published bovid genomes, which were all mapped to the American bison genome (GenBank No. GCA_018282365.1). This analysis resulted in 3,274,372 mapped reads with 0.0759× average genome-wide coverage for CADG467, and 2,365,117 to 749,141,984 mapped reads with 0.2071–35.3602× average genome-wide coverage across the downloaded individuals from ENA (Appendix A). After filtering, we retained 6,000,104 nuclear SNPs for further analyses.

Three newly obtained mitochondrial genomes (i.e., CADG456, CADG465 and CADG467), together with 69 published sequences of the family Bovidae were used to infer a mitochondrial ML phylogenetic tree (Figure 2a). We obtained a topology consistent with previous studies [19,58], in which extant American bison fall within the variability of the extinct steppe bison clade. Similar to the result by Vershinina et al. [17], the bison clade is further divided into three haplogroups, i.e., A, B and C (Figure 2a). Haplogroup A mainly contains extant American bison and extinct steppe bison individuals from North America, whereas haplogroups B and C are extinct lineages without surviving descendants. Haplogroup B was the main lineage of steppe bison, once widely distributed in Eurasia during the Late Pleistocene. Notably, in the ML phylogenetic tree, our three samples do not form a monophyletic clade and fall within two recognized maternal haplogroups. Two of them (CADG456 and CADG467) cluster with high bootstrap support value (100%) with the Holocene Rauchua bison and form the distinct haplogroup C, which is the first lineage that splits from the bison clade. The other sample (CADG465) falls within the mitochondrial diversity of steppe bison haplogroup B (Figure 2a).

To estimate the coalescence time of nodes in the bison mitochondrial phylogeny, we carried out a Bayesian analysis in BEAST based on near-complete mitogenomes, using both root and tip-dating for temporal calibrations. We obtained a topology similar to the ML tree, in which haplogroups A and B diverged at 195 ky BP (95% HPD: 225-171 ky BP), while the split between haplogroup C and the lineage leading to haplogroups A and B was dated at 326 ky BP (95% HDP: 397-264 ky BP). The TMRCA of haplogroups A, B and C are 167 ky BP (95% HDP: 186-153 ky BP), 151 ky BP (95% HDP: 175-131 ky BP) and 130 ky BP (95% HDP: 172-94 ky BP), respectively (Figure 2b).

To expand the number of available bison individuals, we conducted a phylogenetic analysis using a 639 bp D-loop sequence dataset based on Shapiro et al. [10]. The Bayesian phylogenetic tree divides the bison samples into clades 1, 2, 3, 4 and 5 (Figure 3). The number of clades is different from the three main branch structures suggested by complete mitochondrial genomes, which indicate only three haplogroups in Figure 2. By comparing the two trees (Figure 2a and Figure 3), we found that clades 1, 2 and 4 correspond to haplogroup A. Clade 3 corresponds to haplogroup B, and clade 5 corresponds to haplogroup C. Although too few complete mitochondrial genomes may lead to incomplete details of branches, our two samples (CADG456 and CADG467) together with the Rauchua bison (KR350472) keep clustering together and fall in the newly named clade 5, for which there were no representative samples in Shapiro et al. [10], forming a basal branch in all phylogenetic analyses (Figure 2 and Figure 3).

To investigate the nuclear phylogenetic position of bison carrying mitochondrial haplogroup C, we used the 6,000,104 Single Nucleotide Polymorphisms (SNPs) in the dataset, which contains our single sample (CADG467), as well as one Siberian and one Yukon steppe bison, two yaks, three aurochs, six American bison and ten European bison. The nuclear phylogenetic analysis also supports a basal position of the haplogroup C individual within the American/steppe bison clade (Figure 4).

To further explore the genetic affinities of haplogroup C within the extinct steppe bison and extant American bison, we performed a PCA analysis using our available low-coverage nuclear sequence (CADG467) together with nuclear genomes from two steppe bison and six American bison individuals. The result is shown in Figure 5 with 22.5% and 13.2% of the variance explained by PC1 and PC2, respectively. It indicates that the three analyzed steppe bison samples clustered together. However, compared to one steppe bison individual from Yukon (haplogroup A), our sample CADG467 (haplogroup C) shows a closer relationship to a Siberian steppe bison carrying a haplogroup B mitochondrial sequence, indicating a certain phylogeographic structure (Figure 5 and Appendix A).

Additionally, we also performed admixture analyses to investigate the genetic structure of steppe bison, American bison, aurochs, European bison and yak (Figure 5). When K = 3 and 4, five groups were identified, i.e., steppe bison from Siberia, steppe bison and American bison from North America, aurochs, European bison and yak (Figure 6). Additionally, When K = 2–5, a similar genetic structure is found among the steppe bison and American bison individuals, although the proportion of ancestry varies between the samples. CADG467 (bison haplogroup C) shares a higher proportion of ancestry with the bison individual from Siberia (bison haplogroup B) than with the steppe bison from Yukon (bison haplogroup A), while the latter has the same population background as American bison.

## 4. Discussion

ML and Bayesian phylogenetic tree reconstructions using complete mitochondrial genomes indicate that Late Pleistocene fossil bison from Europe, Asia and America group into three major clades (i.e., haplogroups A, B and C), and samples from northeastern China fall into at least two bison lineages. One of our samples (CADG465) clusters into the main Eurasian bison mitochondrial clade (haplogroup B) (Figure 2a), which is consistent with the findings of Shapiro et al. [10]. Haplogroup C, a deeply divergent maternal bison lineage, previously represented by only a single sample from Rauchua, Far Eastern Siberia [14,17,19], is confirmed by two samples (CADG456, CADG467) in this study (Figure 2a and Figure 3). Considering the overlapping radiocarbon dates of the available samples from northeastern China (Figure 2, Figure 3 and Figure 4, Appendix A), the results indicate that the identified two bison evolutionary haplogroups (i.e., haplogroups B and C) coexisted in northeastern China during the Late Pleistocene. Our findings thus indicate a more complex evolutionary scenario of fossil bison in northeastern China than previously thought.

The confirmation of the basal mitochondrial lineage C in this study (Figure 2, Figure 3 and Figure 4) is strikingly similar to the aurochs from northeastern China, which were assigned as an unidentified haplogroup, with a basal position in the auroch’s tree [59]. In addition, it is also true for extinct cave hyenas excavated from the same region, which form a deeply diverging mitochondrial haplogroup of *Crocuta* [33]. It should be noted though, that for nuclear DNA, Chinese cave hyenas form a clade with other Eurasian cave hyenas to the exclusion of African spotted hyenas, whereas in the bison, the clade C individual, for which we could obtain sufficient nuclear data also forms a basal lineage in the nuclear phylogenetic reconstruction (Figure 4). These results suggest that at least some members of Pleistocene fauna in northeastern China are genetically different from their counterparts from other regions and may indicate a long-term ecological refugium for these species in this region. In future studies, mammalian material from this region should receive particular attention to better understand the evolutionary histories of Pleistocene fauna.

Until now, fossil bison carrying haplogroup C were only identified from two isolated sites. Since it was first identified by an Early Holocene sample from Rauchua [19], Vershinina et al. [17] hypothesized that populations carrying haplogroup C possibly persisted south of Rauchua during the Pleistocene. In this study, we detected a Late Pleistocene presence of this haplogroup in northeastern China (Figure 2 and Figure 3), which is indeed south of the location where it was first discovered. In addition, investigation of numerous bison samples from Eurasia and the American continent did not detect additional haplogroup C individuals [10,14,15,18,60,61]. Therefore, haplogroup C might represent a relatively isolated lineage compared to haplogroup B, but its past geographical distribution range still remains unclear due to the very limited number of fossils that were so far found to carry haplogroup C.

The origin of bison haplogroup C has attracted attention since it was identified by ancient DNA analyses [19]. It has been argued that the last surviving Holocene steppe bison population, represented by the Rauchua bison (haplogroup C) “was not a direct descendant of the major population that dominated Siberia during MIS3 and MIS2” [1]. Our phylogenetic time-tree analysis revealed that haplogroup C separated from the main bison clade first, at around 326 ky BP (95% HPD: 397-264 ky BP) (Figure 2b), and the TMRCA of haplogroup C was estimated at 130 ky BP (95% HDP: 172-94 ky BP), indicating that this haplogroup represents an ancient bison lineage, which appeared no later than the late Middle Pleistocene. Given this time scale, it remains an open question not only where it originated, but also, where the population carrying it survived for several hundreds of thousands of years.

However, our admixture analyses, albeit based on limited data, suggest a relatively high similarity between Siberian steppe bison carrying haplogroup B mitogenomes and our Chinese haplogroup C specimen. The PCA also suggests that the analyzed samples of the genus *Bison* show at least some phylogeographical structure (Figure 6), since the two steppe bison samples from Siberia (haplogroup B) and northeastern China (haplogroup C) fall closely together, whereas one steppe bison from Yukon (haplogroup A), for which enough nuclear DNA data are available, is genetically more distant from the analyzed steppe bison samples from Asia.

This finding not only contradicts the results of the maternal phylogenetic trees (Figure 2 and Figure 3), it also questions the interpretation that haplogroup C represents a distinct population as previously suggested [1]. Indeed, it has been pointed out before that caution is warranted in the interpretation of single genetic loci, as the stochastic nature of the genealogical process can affect inference from such loci [62]. Clearly, high-quality whole paleogenome sequences from a sufficient number of geographically and temporally well-spread steppe bison samples will be needed to get a better understanding of the evolutionary history of this widespread Pleistocene species.

## Figures and Tables

**Figure 1 genes-13-01684-f001:**
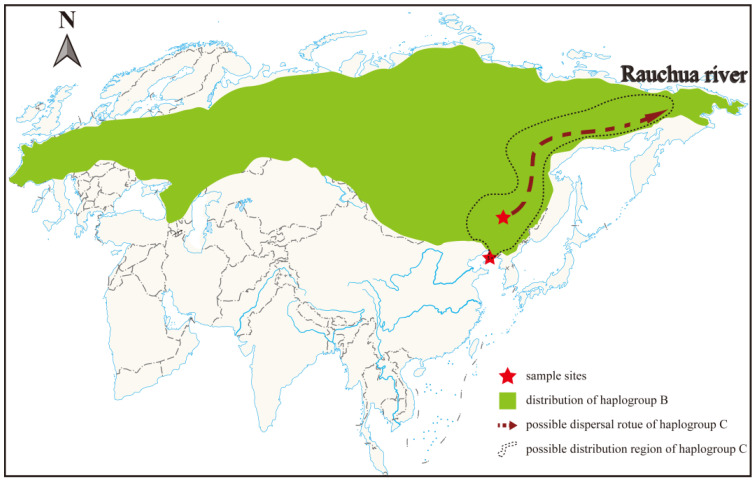
Sampling sites and geographical distribution of steppe bison haplogroups B and C in Eurasia.

**Figure 2 genes-13-01684-f002:**
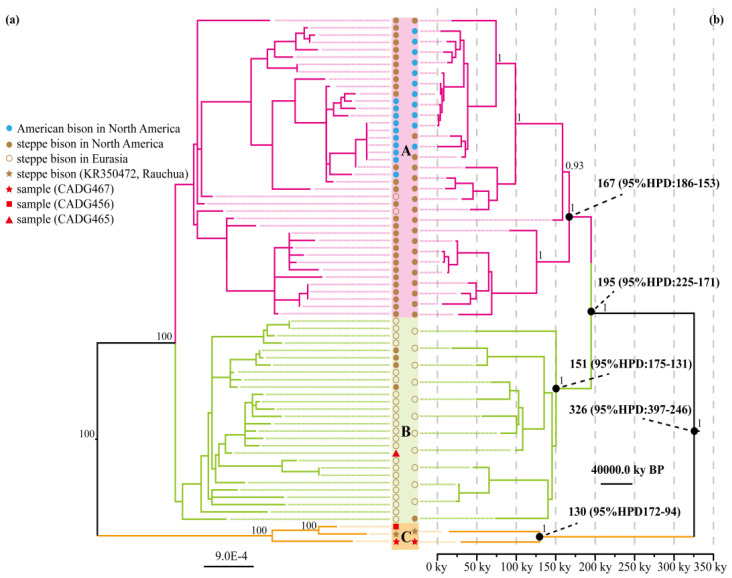
Phylogenetic analyses of bison based on complete mitochondrial genome sequences. (**a**) Maximum-likelihood (ML) phylogenetic tree. Branch labels show bootstrap values derived from 500 replications. (**b**) Time-calibrated Bayesian phylogenetic tree. Numbers above the nodes represent the posterior values. A timescale is placed along the bottom of the tree to facilitate interpretation of tip ages and divergence times. The major divergence events are indicated by black dots and the corresponding divergence times are the median posterior age estimates with 95% credibility intervals.

**Figure 3 genes-13-01684-f003:**
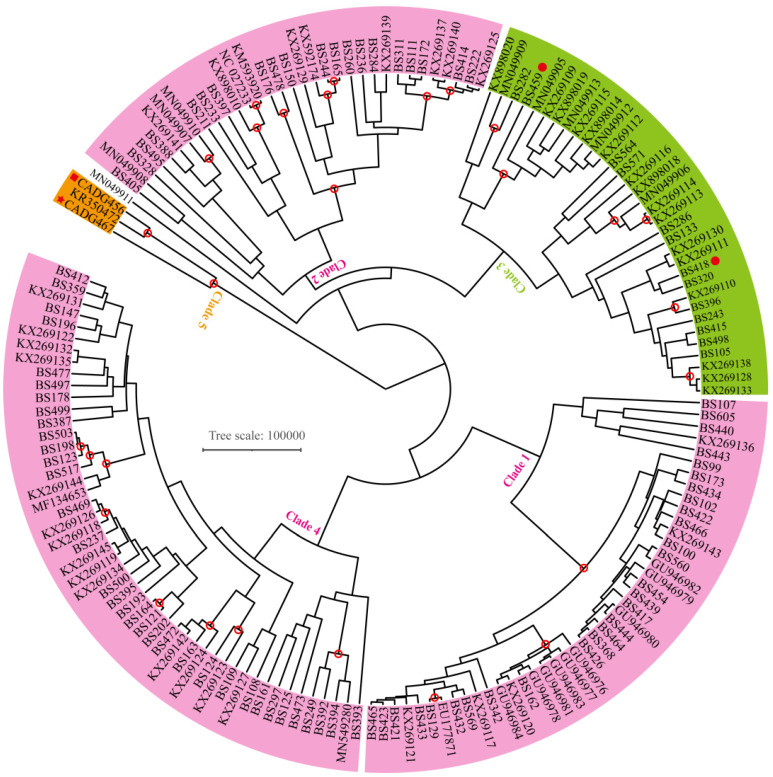
Bayesian phylogenetic tree of bison mitochondrial DNA based on partial D-loop region. Red hollow circles denote nodes supported by posterior probabilities over 0.7. Clade numbers follow Shapiro et al. [10]. The obtained D-loop sequence of CADG465 is too short, so we excluded it from this analysis. Our other two ancient bison samples (CADG467 and CADG456) are shown by a red star and a red square. Two bison samples mark by red solid circles (BS418 and BS459) were collected from northeast China by Shapiro et al. [10]; their median ages were 26,560 and 47,700 cal BP, respectively.

**Figure 4 genes-13-01684-f004:**
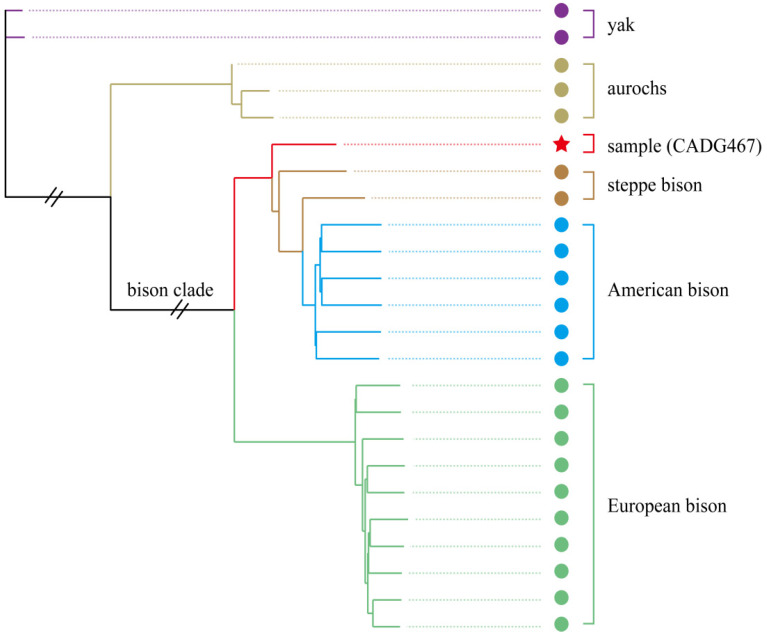
Neighbor-joining phylogeny of bison based on 6,000,104 SNPs using yak as outgroup.

**Figure 5 genes-13-01684-f005:**
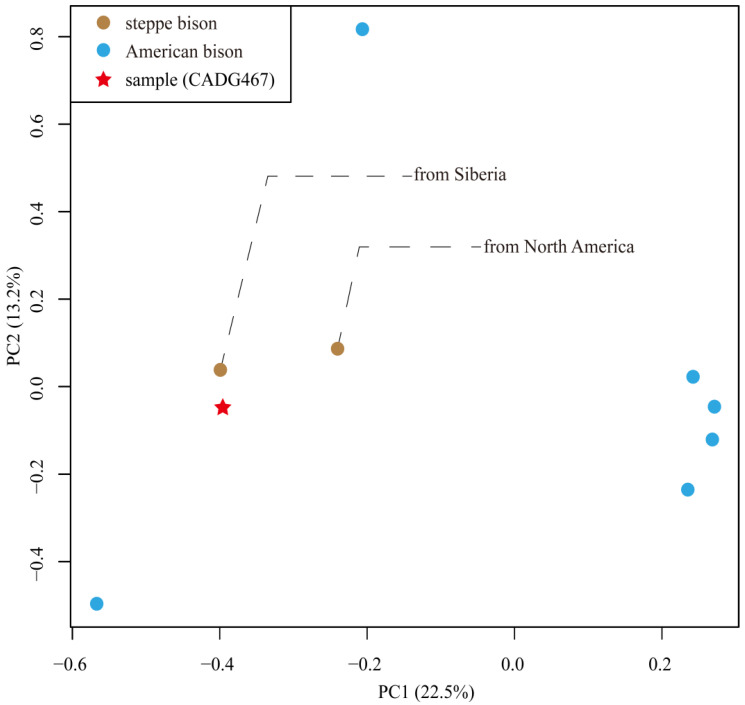
Principal component analysis of bison using 1,137,592 SNPs from our one sample (CADG467), two steppe bison and six American bison. Axis labels indicate the percentage of variance explained by each component. Symbols for each lineage are indicated in the key at the top left.

**Figure 6 genes-13-01684-f006:**
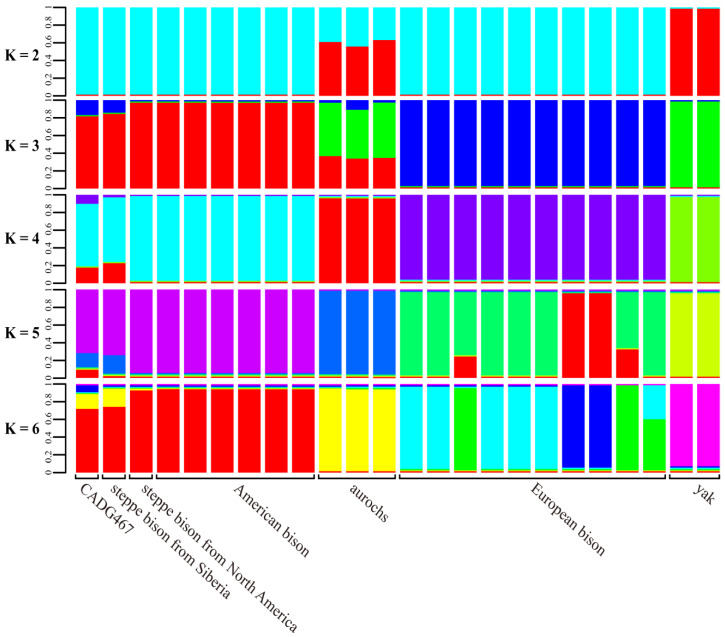
Genetic structure of extinct and extant bison, aurochs and yak using admixture analysis (K = 2–6).

## Data Availability

Two ancient mitochondrial genomes can be found under the GenBank Accession codes OL741537 (CADG456) and OL741538 (CADG467), The address is as follows: GenBank www.ncbi.nlm.nih.gov/genbank, accessed on 10 January 2022.

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
