# Peer review of "Paleogenomes Reveal a Complex Evolutionary History of Late Pleistocene Bison in Northeastern China"

_genes, 2022, doi:10.3390/genes13101684_

Round 1

Reviewer 1 Report

This papers provides a critical insight into ancient DNA of the Bovidae. It provides a critical characterization of bison from ancient ancestors. It is generally a nice read for which the authors are congratulated.

Author Response

Thanks for the reviewer for providing all the helpful suggestions. We have revised the manuscript, and the detailed information is shown in the uploaded file.

Reviewer 2 Report

This interesting and well-written paper describes a complex evolutionary history of Pleistocene bison in northeastern China. The title is descriptive and the abstract corresponds to the main text of the paper. The manuscript is well constructed and is potentially interesting account of the obtained results. The experiments are well conducted and analysis is described in detail. The results here presented an important data of genetic diversity of Pleistocene bison that, as authors concluded, was probably even higher than previously thought, and that northeastern Chinese populations of several mammalian species, including Pleistocene bison, were genetically distinct. Discussion is well reasoning and conclusions are well presented. In this regard, I believe that this particular work provides a theoretical platform for a better understanding of the complex history of fossil bison lineages in northeastern China. The only thing I would recommend to the authors is to carefully check all misspelling in the text.

Author Response

(The authors gave the same response as above.)
